# Paper-Based Detection Device for Microenvironment Examination: Measuring Neurotransmitters and Cytokines in the Mice Acupoint

**DOI:** 10.3390/cells11182869

**Published:** 2022-09-14

**Authors:** I-Han Hsiao, Hsien-Yin Liao, Chao-Min Cheng, Chia-Ming Yen, Yi-Wen Lin

**Affiliations:** 1Department of Neurosurgery, China Medical University Hospital, Taichung 404332, Taiwan; 2School of Post-Baccalaureate Chinese Medicine, College of Chinese Medicine, China Medical University, Taichung 404333, Taiwan; 3Institute of Biomedical Engineering, National Tsing Hua University, Hsinchu 30013, Taiwan; 4Department of Anesthesiology, Taichung Tzu Chi Hospital, Buddhist Tzu Chi Medical Foundation, Taichung 42743, Taiwan; 5School of Post-Baccalaureate Chinese Medicine, Tzu Chi University, Hualien 97004, Taiwan; 6Graduate Institute of Acupuncture Science, College of Chinese Medicine, China Medical University, Taichung 404333, Taiwan

**Keywords:** paper-based ELISA, electroacupuncture, acupoint microenvironment, TRPV1, dorsal root ganglion, somatosensory cortex

## Abstract

(1) Background: The medical practice of acupuncture involves the insertion of a specialized stainless needle into a specific body point, often called an acupoint, to initiate a perceived phenomenon of de-qi sensation. Therefore, the term “de-qi” describes bodily sensations experienced by the recipient during acupuncture, which may include feelings of soreness, heaviness, fullness, numbness, and migration. However, while acupuncture treatments reportedly result in acupoint activation and an increased release of neurotransmitters or cytokines, detecting these substances released into the acupoint microenvironment is often missed or delayed in clinical and basic practice. (2) Methods: To address this situation, we employed a paper-based enzyme-linked immunosorbent assay method to examine acupoint environmental changes using minute volumes of easily accessible acupoint fluids. (3) Results: Our results indicated that while levels of adenosine triphosphate (ATP), interleukin-1β, interleukin-6, glutamate, substance P, and histamine were all increased in the experimental group following electroacupuncture (EA) treatment, contrary results were observed in the sham EA and transient receptor potential vanilloid 1 (*Trpv1*^−/−^) groups. Subsequently, TRPV1 and its associated molecules were augmented in mouse dorsal root ganglion, spinal cord, thalamus, and the somatosensory cortex, then examined by Western blotting and immunofluorescence techniques. Investigations revealed that these elevations were still unobserved in the sham EA or EA in the *Trpv1*^−/−^ groups. Furthermore, results showed that while administering ATP could mimic EA function, it could be reversed by the ATP P2 receptor antagonist, suramin. (4) Conclusions: Our data provide novel information, indicating that changes in neurotransmitter and cytokine levels can offer insight into acupuncture mechanisms and clinical targeting.

## 1. Introduction

Acupuncture originates from Chinese medicine and has a well-documented therapeutic effect on pain management, going back more than three millennia. In ancient literature, acupuncture was recorded as the therapeutic modality of choice for various conditions, especially pain management. Although it was originally conducted using sharp stones, the practice gradually developed to use fine needles, as employed in clinical practice today. The practice of acupuncture involves inserting a fine steel needle into a specific point, often called an acupoint, to initiate a perceived phenomenon of sensation, known as de-qi. Accordingly, the term “de-qi” has been used to describe bodily sensations experienced by the recipient during acupuncture therapy and may include sensations of soreness, heaviness, fullness, numbness, and warmth [1,2]. A study reported that the body contains more than 360 acupoints on 14 body-encompassing meridians. Therefore, acupoints are subsurface body points with abundant sensory nerve terminals, suggesting a strong relationship between them and peripheral sensory afferents [3]. Studies have reported that since acupuncture activates peripheral A and C-fibers to relieve painful sensations, it has been used to treat various diseases and disorders, releasing neurotransmitters, such as ATP, adenosine, dopamine, etc. [4,5]. It has also reportedly increased the release of cytokines to activate receptors for acupuncture signaling [2]. For example, the activation of local receptors can deliver neural impulses to the central spinal cord and then to the brain to modulate the release of neurotransmitters that subsequently drive the vagal–adrenal axis instead of sympathetic reflexes [6].

When an acupuncture needle is inserted into an acupoint in clinical practice, the needle tip goes through the skin and into the muscle layer, leading to associated effects in skin and muscle tissues. Among these associated effects, there was a recent report on its ability to increase the release of adenosine at peripheral sites [4]. A study also reported that needling triggers a spreading increase in purines, including ATP and adenosine, following tissue damage, consistent with an increase in adenosine. ATP released from local acupoints was reported to activate P2 receptors, which can excite nerve terminals for acupuncture analgesia [2]. Simultaneously, the anti-nociceptive effects of the adenosine A1 receptor in both the peripheral and central regions are well documented. As a result, a study reported that they have dramatic side effects, particularly in the heart tissue [7]. The adenosine A1 receptor agonist can also reduce acute and chronic pain [8]. Therefore, Goldmann et al. suggested that acupuncture can reliably increase both ATP and adenosine at peripheral sites to attenuate inflammatory and neuropathic pain [4]. The aforementioned mechanisms indicate that the adenosine A1 receptor is crucial for acupuncture-associated analgesia. Furthermore, a study reported that electroacupuncture at the sciatic nerve could control systemic inflammation by inducing the vagal activation of dopamine release [5]. Consequently, cytokines, such as IL-1β, IL-6, IL-8, and TNF-α, are released through local acupoint activation, regulating the neuroendocrine system. This process also results in acupoint network formation, inducing related neuronal excitation to initiate changes in the neuroendocrine-immune system network called the meridian network [2]. Similarly, inside the acupoint, mast cells, neutrophils, fibroblasts, and keratinocytes have also been reported to release neurotransmitters, such as glutamate, substance P, and histamine. However, the mechanisms accounting for these processes are unclear [2].

Several existing diagnostic methods may be used to detect protein levels, such as Western blotting, neuroimaging, and enzyme-linked immunosorbent assay (ELISA). However, these methods have limited resolution in their current state. Therefore, a low-sample protein detection methodology remains an unmet need, especially one that can be used in an acupoint microenvironment. This need may be fulfilled by employing a paper-based ELISA (p-ELISA) device that is precise, effective, inexpensive to manufacture and operate, and can detect proteins in small volumes [9]. P-ELISA, which relies on ELISA technology performed in paper using an ELISA plate format/design, is thus widely used for biochemical detection. It is fabricated by patterning a hydrophobic polymer in hydrophilic paper to create a platform for protein detection in a microenvironment. Subsequently, it uses specific antibody recognition and high-turnover catalysis by enzymes to provide specificity and sensitivity [10].

It was reported that while acupuncture’s initial primary intent and approach to disease treatment affects the acupoint microenvironment, changes induced when an acupuncture needle is inserted into an acupoint induce ‘de-qi’ and a cascade of other nervous system signaling events. However, the precise mechanisms and interplay of this cascade remain unclear. We previously reported that transient receptor potential vanilloid 1 (TRPV1) is a responding channel for acupuncture. Therefore, this study examines microenvironment changes during electroacupuncture (EA) to predict possible subsequent systemic changes. Using the ST36 acupoint commonly used in clinics to treat several pathological conditions, we further evaluate changes in the level of neurotransmitters and cytokines in mice. The increased molecules then activate TRPV1 and its associated molecules in mouse dorsal root ganglion (DRG), spinal cord (SC), thalamus, and the somatosensory cortex (SSC). Our data indicate that these elevations were still unobserved in the sham EA or EA in the *Trpv1*^−/−^ groups. Administering ATP could mimic EA function and reverse it by the ATP P2 receptor antagonist. Our study provides new evidence regarding the association between EA manipulation in an acupoint and the acupoint microenvironment. This study also provides novel evidence to support the clinical use and mechanisms of EA.

## 2. Materials and Methods

### 2.1. Animals and EA Treatments

The use of mice was approved by the Institute of Animal Care and Use Committee of China Medical University (Permit no. CMUIACUC-2018-247), Taiwan, following the Guide for the Use of Laboratory Animals (National Academy Press, Cambridge, MA, USA). We examined 36 mice, including 27 normal (BioLASCO, Taipei, Taiwan Co., Ltd., Taipei, Taiwan) and 9 Trpv1^−/−^ mice (Jackson Lab, Bar Harbor, ME, USA). Upon receipt, mice were kept under a 12 h light–dark cycle with food and water ad libitum. A sample size comprising nine mice per group was considered the number required for an alpha of 0.05 and a power of 80%. Then, mice were subdivided into four groups: (1) normal mice that did not receive acupuncture (Group 1: Normal), (2) mice receiving 2 Hz EA group (Group 2: EA), (3) mice that received sham EA (Group 3: Sham), and (4) *Trpv1*^−/−^ mice that received 2 Hz EA (Group 4: *Trpv1*^−/−^). The length of the EA treatment was 20 min. Stainless steel acupuncture needles (1 inch, 36G, YU KUANG, Taichung, Taiwan) were bilaterally inserted perpendicularly at a depth of 3 mm into the murine equivalent of the human ST36 acupoint. The murine ST36 is located approximately 4 mm below and 1 mm lateral to the midpoint of the knee in mice. Finally, the number of mice used, and their suffering levels were minimized according to a 3R statement. Laboratory workers were blind to the management allocation during all experiments and analyses. Moreover, the experiment was designed to sacrifice the mice immediately after experimental administration.

### 2.2. p-ELISA

A p-ELISA tool was developed using a wax printer (no. Phaser 8560, Xerox, Norwalk, CT, USA) to manufacture hydrophobic barriers in a 96-well plate fashion on a piece of filter paper (Whatman grade no. 1). Then, several acupuncture-related neurotransmitters or cytokines were immobilized on the p-ELISA filter paper. Horseradish peroxidase (HRP)-conjugated polyclonal rabbit antihuman IgG (DakoCytomation, Glostrup, Denmark) was used as the secondary antibody to label the primary antibody. Subsequently, while phosphate-buffered saline containing 0.1% Tween-20 was used for the washing step, a mixture of 3,3,5,5-tetramethylbenzidine and H_2_O_2_ was used as a coloring reagent to measure HRP oxidization after which the output color signal was recorded using a commercial desktop scanner (no. V370, EPSON, Tokyo, Japan). This p-ELISA method only required a 3 μL sample per well and a short operating time. Furthermore, since the concentrations of different cytokines or neurotransmitters were diverse, we presented the results as a percentage to improve clarity.

### 2.3. Microenvironment Fluid Collection

For p-ELISA analysis of the neurotransmitters and cytokines released into the acupoint microenvironment following acupuncture treatment, we inserted a microdialysis probe one hour before collecting fluid samples. Then, after perfusing the microdialysis probe with Ringer’s solution at 1 μL/min, the probe was inserted perpendicularly at a depth of 3–4 mm into the murine equivalent of the human ST36 acupoint. Next, the probe lysates (40 μL) were collected on ice for 40 min, after which the fluids were immediately used for p-ELISA analysis. Samples were collected before and after the EA treatment protocol. Finally, the perfusion fluids were collected from the microdialysis probe and immediately measured using p-ELISA for ATP, IL-1β, IL-6, glutamate, substance P, and histamine.

### 2.4. Western Blotting

The mice were anesthetized with 5% isoflurane for induction and then underwent cervical dislocation. Subsequently, after the dorsal root ganglion (DRG), spinal column (SC), thalamus, and somatosensory cortex (SSC) tissue samples were immediately dissected to extract proteins, samples were placed on ice and stored at –80 °C. Next, total proteins were homogenized in cold radioimmunoprecipitation (RIPA) lysis buffer containing 50 mM Tris-HCl pH 7.4, 250 mM NaCl, 1% NP-40, 5 mM EDTA, 50 mM NaF, 1 mM Na_3_VO_4_, 0.02% NaN_3_, and 1× protease inhibitor cocktail (AMRESCO), after which the extracted proteins were subjected to 8% sodium dodecyl sulfate-tris glycine gel electrophoresis and transferred to a polyvinylidene difluoride membrane. Next, the membrane was blocked with 5% nonfat milk in TBS-T buffer (10 mM Tris pH 7.5, 100 mM NaCl, 0.1% Tween 20), incubated with a primary antibody in TBS-T with 1% bovine serum albumin (BSA) for one hour at room temperature. The antibodies used were against TRPV1 (∼95 kDa, 1:1000, Alomone, Israel), pPKA (∼40 kDa, 1:1000, Alomone, Israel), pPKC (∼100 kDa, 1:1000, Millipore, Burlington, MA, USA), pPI3K (∼125kDa, 1:1000, Millipore, Burlington, MA, USA), pERK1/2 (∼42–44 kDa, 1:1000, Millipore, Burlington, MA, USA), pAkt (∼60 kDa, 1:1000, Millipore, Burlington, MA, USA), pmTOR (∼60 kDa, 1:500, Millipore, Burlington, MA, USA), and pCREB (∼65 kDa, 1:1000, Millipore, Burlington, MA, USA). However, peroxidase-conjugated anti-rabbit antibody, anti-mouse antibody, and anti-goat antibody (1:5000) were used as the appropriate secondary antibodies. The bands were finally visualized using an enhanced chemiluminescent substrate kit (PIERCE) with LAS-3000 Fujifilm (Fuji Photo Film Co., Ltd., Tokyo, Japan). Where applicable, the image intensities of specific bands were quantified with NIH ImageJ software (Bethesda, MD, USA). β-actin or α-tubulin served as the internal control.

### 2.5. Immunofluorescence

Mice were euthanized with 5% isoflurane via inhalation and cervical dislocation. Then, they were intracardially perfused with normal saline followed by 4% paraformaldehyde. Subsequently, their brain tissue samples were immediately dissected and post-fixed with 4% paraformaldehyde at 4 °C for three days, after which the tissue samples were placed in 30% sucrose for cryoprotection overnight at 4 °C. Next, brain samples were embedded in an optimal cutting temperature compound and rapidly frozen using liquid nitrogen before storing them at −80 °C. After that, the frozen segments were cut into 20-μm widths on a cryostat and instantaneously placed on glass slides, followed by sample fixation with 4% paraformaldehyde, then incubated with a blocking solution comprising 3% BSA, 0.1% Triton X-100, and 0.02% sodium azide, for one hour at room temperature. After blocking, the samples were incubated with a primary antibody (1:200, Alomone), TRPV1, and pERK, prepared in 1% BSA solution at 4 °C overnight. Later, the samples were incubated with a secondary antibody (1:500), 488-conjugated AffiniPure donkey anti-rabbit IgG (H + L), 594-conjugated AffiniPure donkey anti-goat IgG (H + L), or peroxidase-conjugated AffiniPure donkey anti-mouse IgG (H + L), for two hours at room temperature before being fixed with coverslips for immunofluorescence visualization. Samples were finally observed using an epifluorescent microscope (Olympus, BX-51, Tokyo, Japan) with a 20× objective numerical aperture (NA = 1.4), and then images were analyzed using NIH ImageJ software (Bethesda, MD, USA).

### 2.6. Statistical Analyses

Statistical analyses were performed using SPSS 22. The Shapiro–Wilk test was performed to test data normality. P-ELISA data were analyzed by two-way mixed ANOVA. Statistical significance in among all groups was examined using a one-way ANOVA test, followed by post hoc Tukey’s test. Values of *p* < 0.05 were considered statistically significant, and all statistical data are presented as the mean ± standard error (SE).

## 3. Results

### 3.1. EA Increased Cytokines and Neurotransmitters in Mice ST36 Acupoint Microenvironments

Acupuncture needling and electrical stimulation have been reported to initiate a small amount of inflammation [2]. Based on this fact, we sought to examine neurotransmitters and cytokine levels in mouse acupoint microenvironments to assess EA effects using p-ELISA, which is very precise and requires small sample volumes. We observed that ATP levels were similar in normal mice fluids (Figure 1A, n = 9). Additionally, EA treatment significantly increased the level of ATP in the ST36 acupoint microenvironment as measured by p-ELISA (Figure 1A, 124.25% ± 3.13%, n = 9, * *p* < 0.05, F(3, 32) = 49.87). These data validate the reliability of p-ELISA for measuring acupoint microenvironment fluids. We further performed sham EA to verify the specific effect of EA. Investigations revealed that needling without electric stimulation did not alter the level of ATP in normal mice (Figure 1A, 105.35% ± 3.37%, n = 9, *p* > 0.05). We also performed EA on *Trpv1*^−/−^ mice, which were highly reported as EA-responding, and found that EA did not increase the level of ATP in *Trpv1*^−/−^ mice, suggesting a major role in EA manipulation (Figure 1A, 102.02% ± 2.22%, n = 9, * *p* > 0.05). Similar results were also found for IL-1β and IL-6 levels, suggesting their association with EA manipulation (Figure 1B,C, n = 9, * *p* < 0.05, F(3, 32) = 46.02 and 28.43). Subsequently, because EA could trigger the de-qi phenomena, such as soreness, heaviness, fullness, numbness, and migration, we analyzed neurotransmitter levels in the ST36 acupoint microenvironment. Investigations revealed that glutamate, substance P, and histamine increased after EA treatment (Figure 1D–F, n = 9, * *p* < 0.05, F(3, 32) = 14.79, 20.88, and 34.89). However, these phenomena were not observed in experiments using sham EA, suggesting the specific effect of EA (Figure 1D–F, n = 9, * *p* > 0.05). Furthermore, while testing *Trpv1*^−/−^ mice, neurotransmitter augmentation was not observed, indicating that TRPV1 is a key receptor for EA response (Figure 1D–F, n = 9, * *p* > 0.05).

### 3.2. Increased EA-Induced Mediators Augmented TRPV1 and Associated Molecules in the Mice DRG but Were Reduced in Sham EA and Trpv1^−/−^ Mice

EA involves inserting a fine needle into specific acupoints, initiating acupuncture sensation and potential effects. We first examined TRPV1 and related molecules in the peripheral DRG of mice. Then, TRPV1 protein levels and the level of associated molecules in the mice DRG were surveyed to determine their potential acupuncture effects. TRPV1 was expressed and unaltered in normal mice following EA or sham EA manipulation (Figure 2A, black column, n = 6, *p* > 0.05, F(3, 20) = 0.99). Results also showed that TRPV1 was lacking in *Trpv1*^−/−^ mice (Figure 2A, black column, n = 6). Therefore, we subsequently measured the pPKA levels, a downstream molecule of TRPV1, 20 min after EA treatment. A significant increase in pPKA expression levels in the EA group (Figure 2A, red column, n = 6, * *p* < 0.05, F(3, 20) = 14.96) compared with normal mice (Figure 2A, red column, n = 6) was observed. However, an increase in pPKA levels was not observed in the sham EA or *Trpv1*^−/−^groups, demonstrating a specific effect for EA (Figure 2A, blue and green columns, n = 6). Similar results were obtained for pPI3K and pPKC protein levels (Figure 2A, blue and green columns, n = 6, * *p* < 0.05, F(3, 20) = 13.68 and 5.77). We additionally observed that the pAkt and pmTOR proteins, downstream molecules of protein kinases, were significantly potentiated in the EA group (Figure 2B, black and red columns, n = 6) compared with the normal mice group (Figure 2B, black and red columns, n = 6). However, these increases were not shown in sham EA or *Trpv1*^−/−^ groups (Figure 2B, blue and green columns, n = 6). Furthermore, while pERK was expressed in normal mice DRG (Figure 2B, blue column, n = 6), it increased after EA manipulation (Figure 2B, blue column, * *p* < 0.05, F(3, 20) = 8.91, n = 6). Nevertheless, sham EA did not alter the expression of pERK (Figure 2B, blue column, *p* > 0.05, n = 6). Similar results were observed in *Trpv1*^−/−^ mice (Figure 2B, green column, *p* > 0.05, n = 6). Investigations also revealed that while transcriptional factors, such as pCREB, shared the same tendency for increased levels following EA, this effect was lacking in the sham EA or *Trpv1*^−/−^ group (Figure 2B, n = 6).

By immunofluorescence, Western blotting was used to gather quantitative evidence of changes in the DRG (Figure 2C). The immunostaining images of the DRG illustrated similar expression levels of TRPV1 in the normal, EA, and sham EA groups, indicating that EA did not alter the TRPV1 expression during this period. Furthermore, while immuno-positive signals were not observed in the *Trpv1*^−/−^ group (Figure 2C, green signals, n = 3), the EA group demonstrated visibly augmented values and significant increases in pERK protein density compared with the normal group (Figure 2C, red signals, n = 3). However, this increase was not observed in the sham EA or *Trpv1*^−/−^ groups (Figure 2C, red signals, n = 3). Specifically, results also revealed that while the merging of fluorescent signals showed increased co-expression of TRPV1 and pERK in DRG following EA treatment, these co-expression signals were not potentiated in the sham EA and *Trpv1*^−/−^ groups (Figure 2C, yellow signals in merged images, n = 3).

### 3.3. EA Potentiated TRPV1 and Associated Molecules in the Mice SC That Were Not Observed in Sham EA and Trpv1^−/−^ Mice

After EA was performed, SC samples were collected to detect protein modifications based on central transmission. In mice SC (Figure 3), we observed that TRPV1 was not altered in the EA group (Figure 3A, red column, *p* > 0.05, F(3, 20) = 0.04, n = 6) compared with the normal group (Figure 3A, black column, n = 6). Investigations also revealed that while unchanged cells were observed in the sham EA and *Trpv1*^−/−^ groups subjected to the sham operation of EA treatment or the absence of the TRPV1 receptor (Figure 3A, blue and green columns, n = 6), increases were discovered in the density of pPKA, pPI3K, and pPKC, which are the principal pathways for protein kinases (Figure 3A, red, blue, and green columns, * *p* < 0.05, F(3, 20) = 7.01, 8.35, and 13.09, n = 6). However, these increases were reversed in the sham EA or *Trpv1*^−/−^ groups (Figure 3A, red, blue, and green columns, *p* > 0.05, n = 6). Furthermore, crucial downstream molecules in the TRPV1 signaling pathway, pAkt, pmTOR, and pERK, also displayed significant increases in the EA group compared with the normal group (Figure 3B, black and red columns, * *p* < 0.05, F(3, 20) = 10.96, 10.42, and 12.23, n = 6). Similarly, this increase was reversed in the sham EA or *Trpv1*^−/−^ groups (Figure 3B, blue and green columns, *p* > 0.05, n = 6). Results also showed that although the pCREB protein, which was surveyed for its capacity as a transcription factor, was significantly greater in EA-treated mice, significantly lower levels were presented in the other three groups (Figure 3B, n = 6).

Subsequently, we measured immune signals in mice SC to detect changes in the expression of TRPV1 and pERK following EA manipulation. Investigations also revealed that the ratio of TRPV1-positive signals in SC was unaltered in the normal, EA, or sham EA groups (Figure 3C, green fluorescence signals). Additionally, similar to DRG results, TRPV1-positive signals were not observed in *Trpv1*^−/−^ mice. Additionally, results showed that pERK expression in normal mice was enhanced by EA stimulation (Figure 3C, red fluorescence signals). Similar to the findings in the DRG region, the number of pERK-positive signals was dramatically reduced in sham EA and *Trpv1*^−/−^ groups (Figure 3C, red fluorescence, n = 3 mice per group). Moreover, consistent with changes in pERK, qualitatively similar tendencies were observed in merged dual-strained section images of SC (Figure 3C, yellow signals, n = 3 mice per group).

### 3.4. Peripheral EA Treatments Delivered Acupuncture Signals through TRPV1 and Related Molecules in the Mouse Thalamus and SSC

Next, we inspected the DRG and SC, investigating the aforementioned molecules in the mouse thalamus and SSC. Results showed that the expression of TRPV1 was similar in the mice thalamus of normal, EA, and sham groups (Figure 4A, *p* > 0.05, F(3, 20) = 0.11, n = 6). However, TRPV1 expression was absent in the *Trpv1*^−/−^ group compared with all other groups. Moving along the signaling pathway, we analyzed the protein expression levels of pPKA, pPI3K, and pPKC. Although these protein kinases were present in the mice thalamus and elevated after EA treatment, this increase was reversed among the sham EA or *Trpv1*^−/−^ groups (Figure 4A, * *p* < 0.05, F(3, 20) = 7.25, 5.76, and 9.92, n = 6). We also observed a similarity in the expression tendency for pAkt, pmTOR, pERK, and pCREB, whereby a significant increase in the EA group compared with the normal group occurred (Figure 4B, * *p* < 0.05, F(3, 20) = 13.31, 5.12, 11.02, and 8.12, n = 6). Again, the increase was dramatically attenuated in the sham EA and *Trpv1*^−/−^ groups, indicating the specific effect of EA at the ST36 acupoint (Figure 4B, n = 6). The mouse SSC also revealed similar results and tendencies (Figure 5, n = 6).

Finally, qualitative indications in the thalamus were quantified via immunofluorescence. Although immuno-positive signals in the thalamus showed similar levels of existing TRPV1 in normal, EA, and sham EA groups, immuno-positive signals were lacking in *Trpv1*^−/−^ mice (Figure 4C, green fluorescence signals, n = 3). Specifically, while there were significant increases in pERK staining signals in the EA group, in contrast with the normal group, which presented a significant increase in protein signals, normal levels were observed in sham EA and *Trpv1*^−/−^ groups (Figure 4C, red fluorescence signals, n = 3). However, in merged images of dual-strained thalamus sections, qualitatively similar pERK levels were observed (Figure 4C, yellow signals, n = 3 mice per group). Similar results and tendencies were also obtained in mice SSC (Figure 5, n = 3).

### 3.5. Acupoint ATP Injections Mimicked the EA Effect That Was Reversed by ATP P2-Antagonist Suramin

As shown in Figure 6A, while TRPV1 levels in mice SSC remained unaltered in ATP– and ATP+ suramin-injected mice, there was a significant augmentation of pPKA in the ATP-injected group (Figure 6B, red column, * *p* < 0.05, F(2, 15) = 42.69, n = 6) compared with the normal group (Figure 6B, black column). However, the increase in pPKA was attenuated in the ATP + ATP P2-antagonist-suramin injection group (Figure 6B, blue column, n = 6). We further examined the protein levels of pPI3K and pPKC in mice SSC. Our results indicate that although pPI3K and pPKC increased following ATP injections (Figure 6C,D, red column, * *p* < 0.05, F(2, 15) = 18.29 and 13.55, n = 6), these increases were antagonized by suramin injections (Figure 6C,D, red column, *p* < 0.05, n = 6). We further observed that while pAkt and pmTOR protein levels were meaningfully increased in the ATP group (Figure 6E,F, red column, * *p* < 0.05, F(2, 15) = 32.74 and 21.31, n = 6), the potentiation was reliably alleviated in the ATP + suramin group (Figure 6E,F, blue column, n = 6). Similar results were obtained for pERK and pCREB expression (Figure 6G,H, * *p* < 0.05, F(2, 15) = 17.44 and 13.18, n = 6).

## 4. Discussion

Acupuncture is a mechanical stimulation that delivers sensation from the acupoint to systemic organs through meridians. Acupuncture needling initiates a complex release of substances, such as cytokines and neurotransmitters, that activate peripheral nerves. Studies have also reported that increased cytokines or neurotransmitters stimulate the peripheral nerves and transfer signals through the spinal cord to the central nervous system [11,12,13]. According to ancient theory, these signals induce sensations of soreness, heaviness, fullness, numbness, and migration. However, the acupoint microenvironment and relationships among integrated factors remain largely unrevealed. According to modern neuroanatomy studies, the acupoint is a stereoscopic structure with skin, connective tissue, subcutaneous tissue, nerve, and muscle. At each acupoint, peripheral nerve endings, plentiful neuronal or non-neuronal receptors, and capillaries, are highly expressed, which subsequently respond to the stimulation of acupuncture [14,15]. Yet, examining and understanding small changes in the local acupoint microenvironment is an unmet medical problem. Previously, while Goldman et al. reported that acupuncture significantly increased ATP and adenosine at the acupoint [4], Torres-Rosas et al. indicated that dopamine facilitated vagal stimulation of the immune system by EA [5]. Liu et al. also provided neuroanatomical data for the specificity of acupoints in driving specific autonomic pathways [6], whereas Huang et al. demonstrated that mast cells are abundant in acupoints and can be regulated by acupuncture through TRPV1, TRPV2, and TRPV4 channels. As a result, the mast cells initiate mechanical stimuli, triggering acupuncture signals by activating histamine, ATP, or adenosine receptors. Activated neurons project signals to pain-related regions in the spinal cord or higher brain centers for pain relief [16]. Li et al. also concluded that after acupuncture needling, IL-1β and IL-6 were initially released into the acupoint. Sequentially, these cytokines trigger fibroblasts to release ATP and glutamate. After acupuncture treatment, substance P and histamine were reportedly released in mast cells as well. These microenvironment factors can activate representative receptors in the nerve terminals for acupuncture signaling [2]. Luo et al. reported that blood vessels, mast cells, and acetylcholinesterase were present at acupoints. Consequently, EA significantly recruits mast cells near the blood vessels and nerve bundles [16,17]. Furthermore, Wang et al. demonstrated that while mechanical stimuli reliably induced ATP release from human mast cells (HMC-1 cells), the release was highly related to increased intracellular Ca^2+^ [18]. In this study, we noticed that releasing the aforementioned molecules delivered EA signals from the peripheral DRG that passed these sensations to the SC, thalamus, and SSC. However, these signals were not observed in sham EA or *Trpv1*^−/−^ groups, suggesting the specific effect of EA and the crucial role of TRPV1 receptors. These results may be highly correlated with clinical acupuncture analgesia.

Subsequently, this study used p-ELISA to detect microenvironmental changes in the acupoint microenvironment before and after EA treatment. While p-ELISA provides sensitivity and specificity advantages with low sample consumption, convenience, low cost, and ease of use, it is also faster than commercial ELISA because microdialysis fluids can be used for immediate protein detection, and results can be examined with a desktop scanner. Previously, we described how p-ELISA could be used to measure the HIV-1 envelope antigen gp41 in human serum [9]. We also used p-ELISA to detect Escherichia coli with limited detection under 105 colony-forming units [19]. Another publication showed that deep partial-thickness burn blister fluid had higher angiogenin levels, as measured using the p-ELISA method [20]. P-ELISA could also be used for early, preclinical stage Alzheimer’s disease (AD) screening, including for amnestic mild cognitive impairment of AD [10]. Additionally, we further observed that p-ELISA has the advantage of rapidly and conveniently diagnosing disease states when monitoring bullous pemphigoid severity [21]. Therefore, in this study, we used p-ELISA to detect a small volume of microenvironment fluids to examine the effects of acupuncture. We indicated that EA significantly increased the release of ATP, IL-1β, and IL-6. While ATP sequentially triggered mast cells and fibroblasts to release IL-1β, glutamate, substance P, and histamine due to ATP function and suramin antagonism, these cytokines or neurotransmitters then activated individual receptors, having effects of soreness, heaviness, fullness, numbness, and migration. We further indicated that these phenomena were not observed in the sham EA group, suggesting a specific effect of EA on the local acupoint microenvironment. However, these factors were not increased in *Trpv1*^−/−^ mice, suggesting its crucial role in EA treatment.

Recent articles propose that purinergic signaling is involved in the physiological mechanisms responding to acupuncture needling. This hypothesis also suggests that acupuncture needle insertion or electric stimulation induces the release of ATP from keratinocytes or fibroblasts to bind P2 receptors [22,23], providing new insight regarding how acupuncture triggers ATP release to bind its local receptor in an acupoint. Subsequently, ATP was reported to activate the P2×7 receptor and initiate the release of mature IL-1β or IL-6. Specifically, IL-1β is a potent proinflammatory cytokine that responds to several stimuli, including tissue injury. These proinflammatory cytokines are also produced by activated macrophages and further initiate inflammatory responses [24,25].

A recent article indicated that the anti-nociceptive effect of adenosine, which is a product of ATP, was released during acupuncture analgesia [4]. Takano et al. reported the crucial role of ATP and adenosine in acupuncture analgesia through a local increase in adenosine in Human subjects [26]. Kim et al. suggested that the acupoint is a neurogenic inflammatory spot which can initiate the local release of the SP [27]. Our previous article also showed that substance P mediates an analgesic effect in muscle nociceptors [28]. Wang et al. reported that histamine and ATP were released in acupoint and delivered acupuncture signals to the spinal cord to produce acupuncture analgesia [16]. He et al. reported that acupuncture reliably induced the release of ATP through the stimulation of P2×3, P2×4, and P2×7 to induce analgesia. ATP can bind to P2 receptors to activate PKA, PKC, and CREB to relieve pain [29]. Herein, we provided evidence that EA significantly increases the ATP concentration, further activating the secretion of IL-1β, IL-6, and several neurotransmitters. Furthermore, we observed that the administration of ATP mimicked EA function and could be further eliminated by the ATP P2 antagonist, suramin. Finally, the aforementioned factors activated TRPV1 receptors that send EA signals from peripheral DRG to the central SC, thalamus, and SSC, initiating needling sensations, after which we used p-ELISA to detect these factors immediately and suitably represent microenvironment changes. Our results provide clinical implications of these neurotransmitters or cytokines in acupuncture analgesia (Figure 7).

## Figures and Tables

**Figure 1 cells-11-02869-f001:**
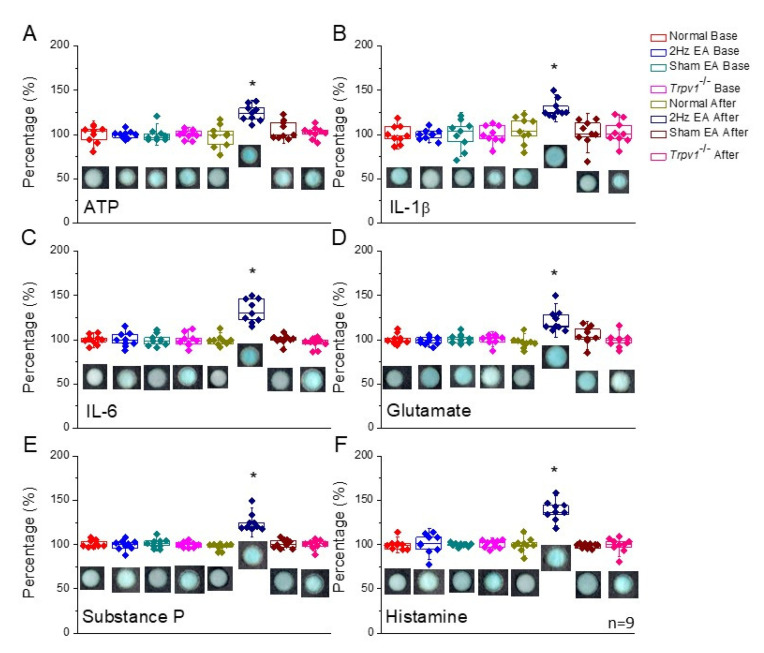
Colorimetric results (intensity) from our p-ELISA test for (**A**) ATP, (**B**) IL-1β, (**C**) IL-6, (**D**) glutamate, (**E**) substance P, and (**F**) histamine. The color intensity is shown as a percentage. Asterisks indicate statistical significance when compared to the base condition. n = 9 in all groups.

**Figure 2 cells-11-02869-f002:**
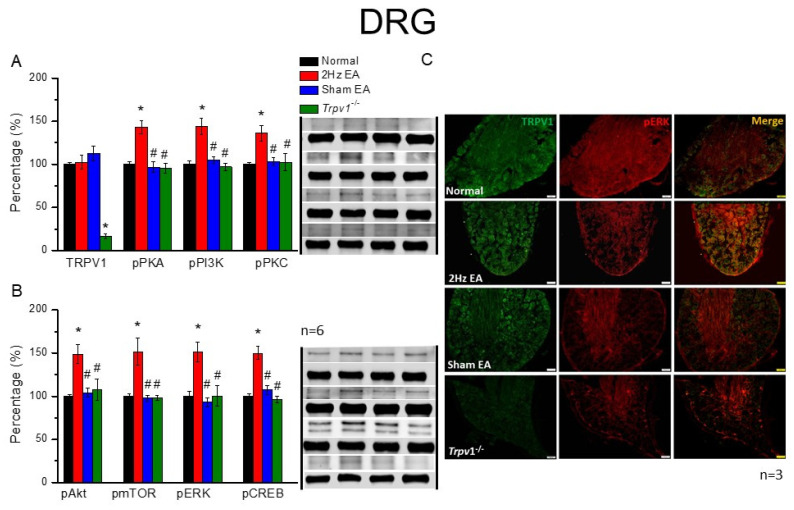
The levels of TRPV1 and related molecules in mice DRG. The immunoblotting bands have four lanes of protein expression associated with Normal, 2 Hz EA, sham EA, and *Trpv1*^−/−^ groups. (**A**) TRPV1, pPKA, pPI3K, and pPKC (**B**) pAkt, pmTOR, pERK, and pCREB protein levels in all group. Asterisks (*) indicate statistical significance when compared with the normal group. Hashtag symbols (#) indicate statistical significance when compared to the EA group. n = 6 in all groups. (**C**) Immunofluorescence staining of TRPV1, pERK, and double staining protein expression in the mice DRG. TRPV1, pERK, and TRPV1/pERK double staining, immuno-positive (green, red, or yellow) signals in the mice DRG region. Scale bar means 50 μm. n = 3 in all groups.

**Figure 3 cells-11-02869-f003:**
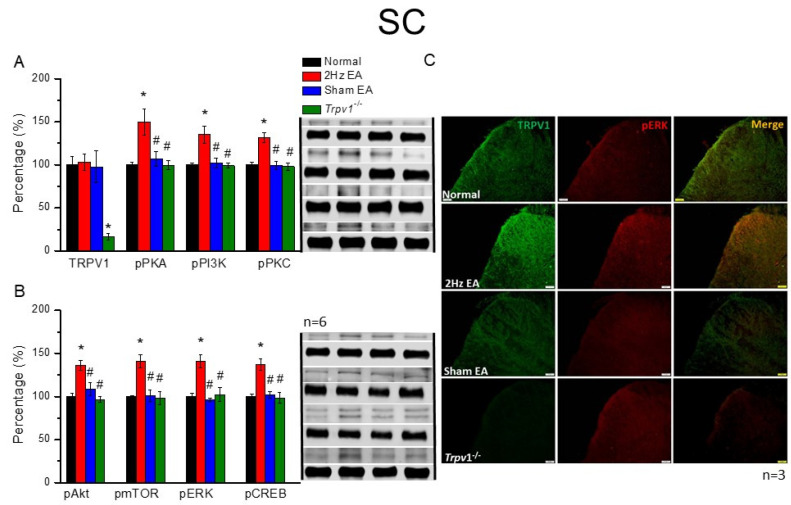
The levels of TRPV1 and related molecules in the mice SC. The immunoblotting bands have four lanes of protein associated with Normal, 2 Hz EA, sham EA, and *Trpv1*^−/−^ groups. (**A**) TRPV1, pPKA, pPI3K, and pPKC (**B**) pAkt, pmTOR, pERK, and pCREB protein levels in all group. Asterisks (*) indicate statistical significance when compared with the normal group. Hashtag symbols (#) indicate statistical significance when compared to the EA group. n = 6 in all groups. (**C**) Immunofluorescence staining of TRPV1, pERK, and double staining protein expression in the mice SC. TRPV1, pERK, and TRPV1/pERK double staining, immuno-positive (green, red, or yellow) signals in the mice SC region. Scale bar means 50 μm. n = 3 in all groups.

**Figure 4 cells-11-02869-f004:**
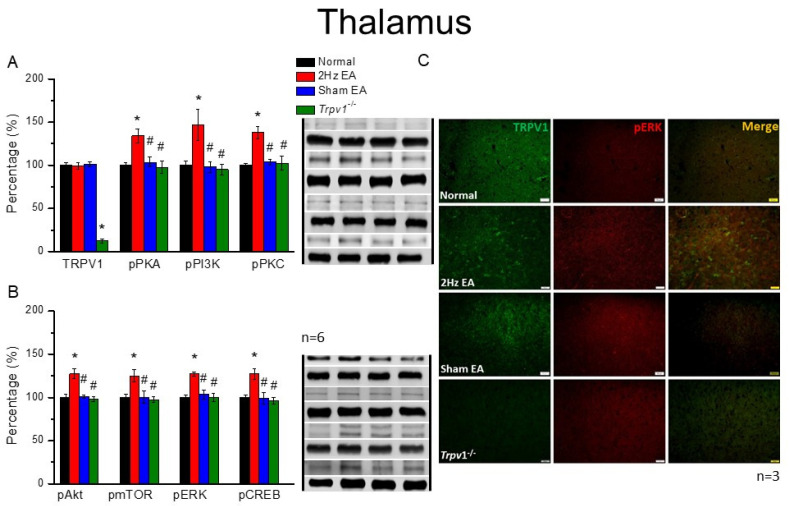
The levels of TRPV1 and related molecules in the mice thalamus. The immunoblotting bands have four lanes of protein expression associated with Normal, 2 Hz EA, sham EA, and *Trpv1*^−/−^ groups. (**A**) TRPV1, pPKA, pPI3K, and pPKC (**B**) pAkt, pmTOR, pERK, and pCREB protein levels in all group. Asterisks (*) indicate statistical significance when compared with the normal group. Hashtag symbols (#) indicate statistical significance when compared to the EA group. n = 6 in all groups. (**C**) Immunofluorescence staining of TRPV1, pERK, and double staining protein expression in the mice thalamus. TRPV1, pERK, and TRPV1/pERK double staining, immuno-positive (green, red, or yellow) signals in the mice thalamus region. Scale bar means 50 μm. n = 3 in all groups.

**Figure 5 cells-11-02869-f005:**
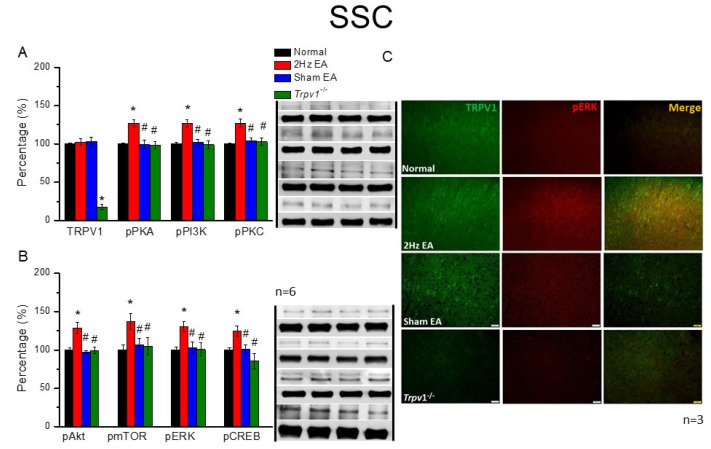
The levels of TRPV1 and related molecules in the mice SSC. The immunoblotting bands have four lanes of protein expression associated with Normal, 2 Hz EA, sham EA, and *Trpv1*^−/−^ groups. (**A**) TRPV1, pPKA, pPI3K, and pPKC (**B**) pAkt, pmTOR, pERK, and pCREB protein levels in all group. Asterisks (*) indicate statistical significance when compared with the normal group. Hashtag symbols (#) indicate statistical significance when compared to the EA group. n = 6 in all groups. (**C**) Immunofluorescence staining of TRPV1, pERK, and double staining protein expression in the mice SSC. TRPV1, pERK, and TRPV1/pERK double staining, immuno-positive (green, red, or yellow) signals in the mice SSC region. Scale bar means 50 μm. n = 3 in all groups.

**Figure 6 cells-11-02869-f006:**
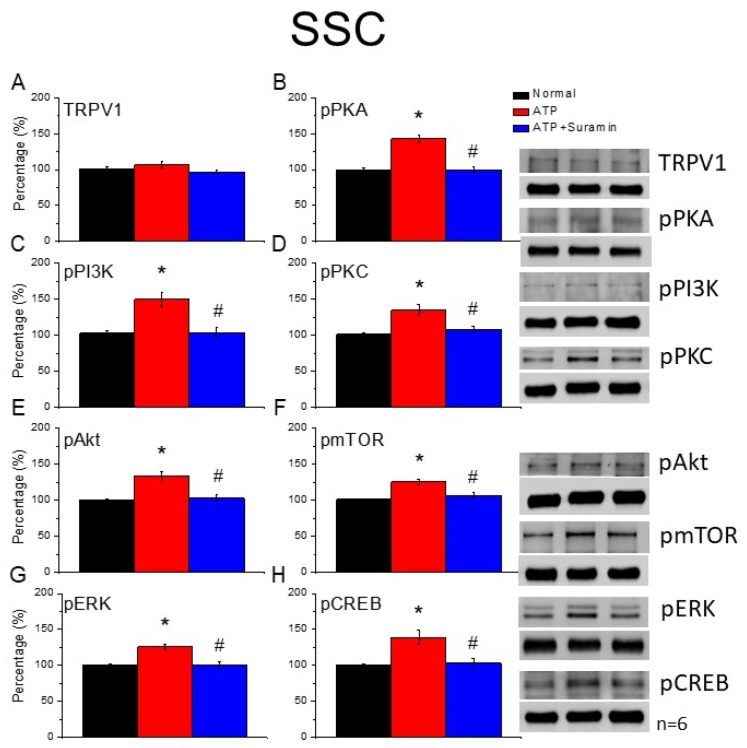
The levels of TRPV1 and associated factors in the mice SSC after ATP injection. The immunoblotting bands have four lanes of protein expression associated with Normal, 2 Hz EA, sham EA, and Trpv1^−/−^ groups. (**A**) TRPV1, (**B**) pPKA, (**C**) pPI3K, (**D**) pPKC, (**E**) pAkt, (**F**) pmTOR, (**G**) pERK, and (**H**) pCREB protein levels in all group. Asterisks (*) indicate statistical significance when compared with the normal group. Hashtag symbols (#) indicate statistical significance when compared to the ATP group. n = 6 in all groups.

**Figure 7 cells-11-02869-f007:**
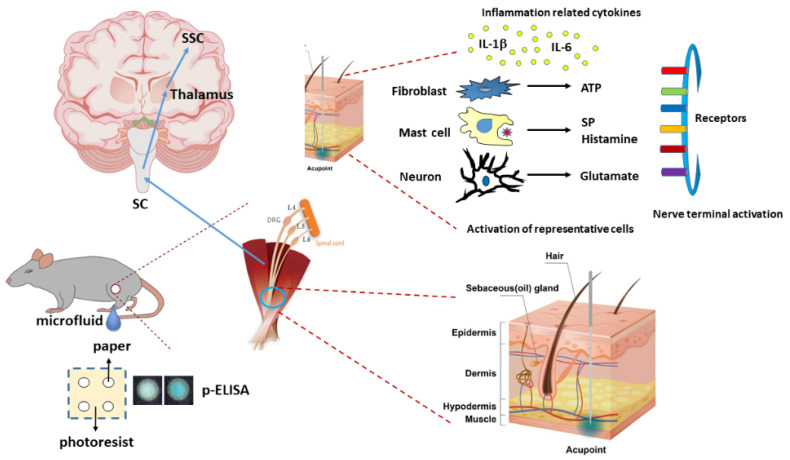
Schematic illustration of microenvironment changes and signaling pathways underlying 2 Hz EA. The summary illustration displays the importance of mechanisms involving cytokines and neurotransmitters in acupoint. EA can directly activate the release of ATP, IL-1β, IL-6, glutamate, substance P, and histamine.

## Data Availability

The datasets supporting the conclusions of this article are included within the article.

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
