# Peer review of "Paper-Based Detection Device for Microenvironment Examination: Measuring Neurotransmitters and Cytokines in the Mice Acupoint"

_cells, 2022, doi:10.3390/cells11182869_

Round 1

Reviewer 1 Report

This work developed a method to examine the neurotransmitters or cytokines releasing into the acupoint microenvironment, which provided novel evidence to support the clinical use and mechanisms of electroacupuncture. But there are some questions that need to be resolved before published.

1.     The p-ELISA analysis is to insert a microdialysis probe into the acupoint to collect fluid samples, so how to obtain the accurate acupoint location, which seems to be important.

2.     For the detection of neurotransmitters or cytokines releasement in the brain, are there any adverse effects of inserting a microdialysis probe into the brain? If so, how to solve this problem?

3.     The sensitivity and accuracy of p-ELISA analysis may be interfered for multi-component detection, and the results should be compared to standard assays.

4.     The description of the abscissa in Figure 3B is not clear.

Reviewer 2 Report

Comments to the Author

The present paper was to examine “Paper-Based Detection Device for Microenvironment Examination: Measuring Neurotransmitters and Cytokines in the Mice Acupoint”. In their review, the authors suggested that the transient receptor potential vanilloid 1 (TRPV1) is the crucial channel for acupuncture. Thus, the author used the ST36 acupoint to treat pathological conditions and then evaluated the level of neurotransmitters and cytokines in mice. The results showed that the 2Hz EA produced increases in these molecules induced TRPV1 and its associated molecules in dorsal root ganglion (DRG), spinal cord (SC), thalamus, and the somatosensory cortex (SSC) compared to the normal group. However, the sham EA and EA in the Trpv1-/- groups did not reveal significant differences in these molecules. Furthermore, ATP administrations could be reversed the effect by the ATP P2 receptor antagonist, suramin. The topic is seemingly attractive for me. It is an interesting topic. However, some issues should be concerned below.

1. Figure 1 used a wrong statistical analysis and draw a wrong figure. For example, the experimental design involved the normal, 2 Hz EZ, sham EZ, and Trpv1 -/- four groups. The data could not be used one-way ANOVA to compared with the base and after groups for normal, 2Hz EA, sham EZ, and Trpv1-/- conditions. The data should be analyzed by a 4 x 2 mixed two-way (groups vs time) ANOVA. Then, one-way ANOVA was performed for group. When appropriate, post hoc with Tukey was conducted for base or after among normal, 2Hz EZ, sham EZ, and Trpv1 -/- groups. This crucial point should be revised.

2. Figures 2-5 have the same deficits for drawing figures and statistical analysis. For example, the figure should be considered for changes. The authors used ANOVA compared with the same molecule (such as Trpv1, pPKA, pPI3K, and pPKC) between different groups. However, these figures were drawn the normal, 2Hz EZ, sham EA, and Trpv1 -/- groups in the x-axis. The molecule (such as Trpv1, pPKA, pPI3K, and pPKC) were drawn in the y-axis. This format cannot reveal the correct results of the statistical analysis. In my opinion, the figures should be revered between x-axis and y-axis. The x-axis should be drawn various molecules (such as Trpv1, pPKA, pPI3K, and pPKC) and the y-axis must be all different groups including normal, 2Hz EZ, sham EA, and Trpv1 -/- groups. Accordingly, one-way ANOVA was respectively conducted for normal, pPKA, pPI3K, and pPKC groups. Then, post hoc with Tukey was performed. Each molecular would be analyzed by one-way ANOVA among all groups. This point should be revised.

3. Figure 6 should be analyzed by one-way ANOVA and then post hoc with Tukey when appropriate. The results of Figures should be written F value besides p value.

4. All figures should be shown the n numbers.

5. All results should be discussed the clinical implications in the Discussion section.

6. The behavioral and molecules data need to be linked with the previous findings and explain its meanings in the Discussion section.

7. Why do the authors choose ATP P2 receptor antagonist to reverse the effect of EA function? It should be clarified in the Introduction section, and the findings of the ATP P2 antagonist should be discussed in the Discussion section with the previous data.

The present study is an interesting topic; however, it cannot be considered for acceptance in the current status. Some major deficits should be revised further.

Round 2

Reviewer 2 Report

Comments to the Author

The authors did not respond to all comments. Moreover, the authors did not respond to all comments with point by point. Their revised manuscript is not easy to catch out with the comments. It is suggested the second round revision should be clearly used the way of point by point for showing their responses. My comments are as follows.

Major.

1. Figure 1 used a wrong statistical analysis and draw a wrong figure. For example, the experimental design involved the normal, 2 Hz EZ, sham EZ, and Trpv1 -/- four groups. The data could not be used one-way ANOVA to compared with the base and after groups for normal, 2Hz EA, sham EZ, and Trpv1-/- conditions. The data should be analyzed by a 4 x 2 mixed two-way (groups vs time) ANOVA. Then, one-way ANOVA was performed for group. When appropriate, post hoc with Tukey was conducted for base or after among normal, 2Hz EZ, sham EZ, and Trpv1 -/- groups. This crucial point should be revised.

My comments: This point did not comprehensively answer. The authors only changed Figure1; however, their statistical analysis was not revised. The present data should be analyzed by two-way ANOVA first.

2. Figures 2-5 have the same deficits for drawing figures and statistical analysis. For example, the figure should be considered for changes. The authors used ANOVA compared with the same molecule (such as Trpv1, pPKA, pPI3K, and pPKC) between different groups. However, these figures were drawn the normal, 2Hz EZ, sham EA, and Trpv1 -/- groups in the x-axis. The molecule (such as Trpv1, pPKA, pPI3K, and pPKC) were drawn in the y-axis. This format cannot reveal the correct results of the statistical analysis. In my opinion, the figures should be revered between x-axis and y-axis. The x-axis should be drawn various molecules (such as Trpv1, pPKA, pPI3K, and pPKC) and the y-axis must be all different groups including normal, 2Hz EZ, sham EA, and Trpv1 -/- groups. Accordingly, one-way ANOVA was respectively conducted for normal, pPKA, pPI3K, and pPKC groups. Then, post hoc with Tukey was performed. Each molecular would be analyzed by one-way ANOVA among all groups. This point should be revised.

My comments: The figures 2-5 have been changed. However, the analysis of one-way ANOVA needs to descript their F value and degree of freedom and its p value for each molecular analysis such as TRPV1, pPKA, pPI3K, and pPKC. The results of Figures 2-5 need to be revised again.

3. Figure 6 should be analyzed by one-way ANOVA and then post hoc with Tukey when appropriate. The results of Figures should be written F value besides p value.

My comments: Please add the degree of freedom for each F value.

4. All figures should be shown the n numbers.

My comments: OK.

5. All results should be discussed the clinical implications in the Discussion section.

My comments: It should be clearly pointed out where the discussion is.

6. The behavioral and molecules data need to be linked with the previous findings and explain its meanings in the Discussion section.

My comment: It is not fully responded to. Where is your revision about this point?

7. Why do the authors choose ATP P2 receptor antagonist to reverse the effect of EA function? It should be clarified in the Introduction section, and the findings of the ATP P2 antagonist should be discussed in the Discussion section with the previous data.

My comment: This point did not be responded to.

The present study is an interesting topic; however, it cannot be considered for acceptance in the current status. Some major deficits should be revised further.

Author Response

Reviewer 2:

  1. My comments: This point did not comprehensively answer. The authors only changed Figure1; however, their statistical analysis was not revised. The present data should be analyzed by two-way ANOVA first.

Response: Thanks for the comment. We revised the statistical analysis. All data in figure 1 were analyzed by a 4 x 2 mixed two-way (groups vs time) ANOVA. The F value was shown besides p value. We also added the statement in the revised “Statistical analyses” (Page 5, highlighted in yellow).

  1. My comments: The figures 2-5 have been changed. However, the analysis of one-way ANOVA needs to descript their F value and degree of freedom and its p value for each molecular analysis such as TRPV1, pPKA, pPI3K, and pPKC. The results of Figures 2-5 need to be revised again.

Response: Thanks for the comment. We descript the F value and degree of freedom and its p value for each molecular analysis such as TRPV1, pPKA, pPI3K, and pPKC. The results of Figures 2-5 were revised again (Pages 6-9, highlighted in yellow).

  1. Figure 6 should be analyzed by one-way ANOVA and then post hoc with Tukey when appropriate. The results of Figures should be written F value besides p value.

My comments: Please add the degree of freedom for each F value.

Response: Thanks for the comment. We added the degree of freedom for each F value besides p value (Page 10, highlighted in yellow).

  1. All results should be discussed the clinical implications in the Discussion section.

My comments: It should be clearly pointed out where the discussion is.

Response: All results were discussed the clinical implications in the Discussion section (Pages 11 and 12, highlighted in green).

  1. The behavioral and molecules data need to be linked with the previous findings and explain its meanings in the Discussion section.

My comment: It is not fully responded to. Where is your revision about this point?

Response: The behavioral and molecules data were linked with the previous findings and explain its meanings in the Discussion section (Pages 12 and 13, highlighted in red).

  1. Why do the authors choose ATP P2 receptor antagonist to reverse the effect of EA function? It should be clarified in the Introduction section, and the findings of the ATP P2 antagonist should be discussed in the Discussion section with the previous data.

My comment: This point did not be responded to.

Response: We choose ATP P2 receptor antagonist to reverse the effect of ATP function according to recent articles. We clarified it in the revised Introduction section, and the findings of the ATP P2 antagonist was discussed in the Discussion section with the previous data. (Pages 2, 12 and 13, highlighted in purple).